# Deprived Muslims and Salafism: An Ethnographic Study of the Salafi Movement in Pekanbaru, Indonesia †

**Andri Rosadi**

Department of Sociology, School of Social Sciences and Humanities, Universitas Islam Negeri Sunan Kalijaga, Yogyakarta 55281, Indonesia; andrirosadi@uinib.ac.id
† This article is part of my thesis chapter submitted to the School of Social Sciences and Psychology, Western Sydney University in 2019.

**Abstract:** This article analyses the process of reversion to Salafism in Pekanbaru, Indonesia in the context of Muslims who have returned to Islam as a solution to their sense of deprivation. This return to Islam is considered by many as an initial solution to a feeling of deprivation which often manifests itself as a form of spiritual 'emptiness', accompanied by anxiety, depression and a lack of direction in life. The analysis in this article is based on extensive reading of relevant literature, participatory observation, and interviews conducted during fieldwork in Pekanbaru from July 2015 to June 2016. The discussion is based on three case studies of Salafi members, detailing their reversion to Salafism and the personal and sociological reasons for their choice to return to Islam, i.e., Salafism, after a certain period of time in their lives. Findings show that those who join the Salafi movement have previously experienced relative deprivation which led to a sense of existential deprivation.

**Keywords:** Salafism; reversion; deprivation

## 1. Introduction

The role of Salafism in Indonesian society is gradually increasing. However, it remains a neglected subject of research, particularly in marginal areas of Indonesia. This article explores the phenomenon of Islamic revivalism in Pekanbaru, Sumatra, in terms of how the Salafi members deal with relative deprivation. A study of Sumatran Salafism is required as most recent studies on Indonesian Salafism have focused on Java, with scant reference to other regions of Indonesia.

The objective of this study is to explore the role of Salafism in how lapsed or non-religious Muslims in Pekanbaru seek to overcome the suffering of existential deprivation by adopting Salafism and joining the Salafi community. This aim was approached by focusing on the reasons why people revert to Islam by choosing only Salafism among the numerous other Islamic movements which exist in Pekanbaru.

In this study, I used the relative deprivation theory, enabling me to reveal socio-cultural factors which caused non-religious Malays to revert to Islam. This theory was selected to gain a more comprehensive understanding of Salafism in Pekanbaru. Salafism operates as a revivalist movement, acting to overcome the problems faced by those struggling with existential deprivation associated with modern life. It is an attempt to return to the teachings of the Qur'an and *sunnah* (sayings and deeds of the Prophet Muhammad), exemplified by the *al-salaf al-salih* (pious predecessors comprising the first three generations of Muslims).

The study of the Salafi movement in Pekanbaru is part of the effort to understand the religious changes occurring in Sumatra. The religious changes on an individual level can take two forms: first, within a given religious tradition (reversion), and second, between religions, (conversion). The former is manifested in a change of religious affiliation, preferences and participation, while the latter may involve conversion from one religion to

another, for example from Christianity to Islam or vice versa. Since the context of this study is the Muslim society of Pekanbaru, where Salafism is very active, my focus is on reversion. Moreover, the use of the term conversion to refer to religious change within one religion is considered incorrect and is uncommon in scholarly literature. The correct term, reversion, is therefore used.

## 2. Methodology

This study utilised case studies and the ethnographic method. According to Benjamin (2005), ethnography may denote a methodology that mainly focuses on the social and cultural contexts of the observed cultures. This method is suited to address themes of research related to perspective, world-views, social interaction and identity (Marvasti 2004). Religion has much to do with the way people see their world, as well as how they establish relationships with others as social beings. Thus, this method is best suited to the study of Salafism as a religious movement in a specific social context in Pekanbaru, Indonesia.

Data was collected during fieldwork through different methods. Participant observation was conducted in Pekanbaru from July 2015 to June 2016. This was important as it enabled me to observe, in person, various events in the research area. Moreover, I sometimes found that a particular description given by an informant during an interview required an observation to validate it. For example, the information given by a participant that the Salafis have strong solidarity among themselves could be validated by observing their attitudes and behaviour in religious rituals and social events they regularly conduct. Similarly, how far a Salafi is attached to the Salafi movement can be best seen from their involvement in Salafi community activities. In these cases, observation was an effective way to address the limitations of the interview method. I conducted observation for 12 months across various locations and on various occasions in Pekanbaru, including in the central part of the city and the suburban areas of Rumbai and Kulim. Unlike interviewing, which to some extent has lost its natural setting (Fontana and Frey 2008), the participant observation method is an effective way to listen to, and capture, authentic conversation among Salafis in a natural setting. During the fieldwork, I spent significant time in various social settings including *aqiqa* (Islamic welcoming ceremony for a new baby), chatting with some Salafi informants casually at restaurants, attending their barbeque parties and regular main prayers and sermons in the Salafi mosques. These allowed me to obtain substantial details about the intentions, beliefs, attitudes and behaviour of the Salafis. I also visited some Salafi schools, such as Imam Shafii Primary School and Ummu Sulaym Islamic Boarding School, to talk with the teachers and principals in their offices.

The second method of collecting data was through in-depth interviews. These can yield substantial details, which is useful if there is scant textual information on the subject (Howe and Lewis 1993). Additionally, the method provides opportunities to delve deep into the interviewee's personal behaviour and experience (Marvasti 2004). Interviews in this study generated extensive verbal data which was later converted into a written form and personal life stories of Salafi members. The interviewees can be termed competent members (Blasi and Weigert 1976), as they are considered representatives of their group (Babbie 2008). It is worth noting here that the Salafis strongly uphold the belief that the interaction between unmarried non-*mahram* (with whom one is not related by blood and, therefore, can marry) males and females is not allowed by Islam. For this reason, the informants were male Salafis and though not intentionally chosen, they were all middle-aged, ranging from 35 to 45 years old.

During the in-depth interviews, I used semi-directive and semi-structured questions to obtain information from the informants. The in-depth interviews were conducted face to face in locations such as mosques and coffee shops decided by the interviewees. Interviews were recorded only if the participant had given consent to do so. I asked participants about their life histories related to the "turning point" in their lives that led them to Salafism. In so doing I asked them about their socio-cultural background, the reference group which influenced them in their choice of Salafism, the reasons behind their choice and their role

and expectations about joining the Salafi movement. The interviews were conducted in the Indonesian language. The recorded interviews were transcribed, while for unrecorded ones, written notes were taken during the interview. Only the relevant parts of the interviews were translated into English, which appear as direct citations in this article. All informants were clearly informed about the purpose of the research. They were also asked to provide their written consent before conducting the interviews. Some of them gave consent to use their real names, while others requested that I use pseudonyms. However, to protect all the informants' privacy, all names in this article are pseudonyms.

Library research represents the third and final method of data collection. The media I accessed included written documents, such as school curricula, brochures and textbooks, or recorded activities, such as CDs or cassettes containing preaching and teaching materials.

Unit analysis of this study is based on individuals, characterised by gender (male Salafis), age (from 35 to 45 years old) and attitudes (religious and non-religious). In general, there are three stages of data analysis, namely data reduction, data display, and conclusions (Berg 2004).

Pekanbaru, the research scene, is the capital city of Riau Province, Indonesia. It is geographically located in the middle of Sumatra Island, close to Malacca Strait. Recently, the city has undergone rapid development, propelled by growth in the petroleum, plantation, pulp and paper industries and service and commerce sectors. The strong economic growth can be seen in the establishment of modern markets and entertainment centers. In addition to the economic growth, urban sprawl and migrant inflows from the neighbouring provinces, particularly West Sumatra, North Sumatra and as far as Java, increase every year which is altering the demographic composition in Pekanbaru (Pekanbaru 2015).

Findings show that those who join the Salafi movement have previously experienced relative deprivation which has led to a sense of existential deprivation. In contrast to a perceived lack of life's necessities in the case of deprivation, the emptiness and lacking in existential deprivation are felt in a religious or spiritual sense, and manifest in feelings of anxiety, emptiness and a sense of a lack of meaning in their life. It is these feelings which motivate them to search for the true and pure interpretation of Islam, in an attempt to quench their thirst for spiritual understanding and comfort. In many cases, the process of adaptation to Salafism is not linear in nature and is accompanied by doubt, contemplation, and for some informants, resistance from their families. Despite these challenges, the participants interviewed stated they felt content and fulfilled once they had found Salafism. This will be discussed in detail below, where it is argued that both social and normative factors, such as the Salafi movement's strong solidarity and purity attract people to join. The discussion is based on three case studies of Salafi members, detailing their reversion, i.e., the personal and sociological reasons for their choice to return to Islam after a certain period of time in their lives of not practicing religion.

### 3. Salafism: A Contested Concept

Salafism is a term widely used to refer to those Muslims who call for the return to the teachings and practices of their pious predecessors (*al-salaf al-salih*). Practically, it is an umbrella concept that incorporates a wide range of different Muslim communities aiming at purifying Islamic teachings and practices. Etymologically, *Salaf* means time past or what happened in the past (al Thalibi 2006; Ma'luf 1986), however over centuries, the concept of Salafism has transformed. It is now attributed to a major Islamic current emerging in Saudi Arabia, aiming to purify Islamic teachings and practices (Adonis 1998; al Buthy 1998). Salafis are also known by some scholars as Wahhabis (Ayoob 2009; Commins 2006; Sirozi 2005). The majority of funding for international Salafi activity comes from Saudi Arabia (Haghayegi 2002), which according to Gadzey (2005) provides funds for the operation of 1500 mosques, 210 Islamic centers and almost 2000 colleges in various countries where Wahhabism is taught to local people.

Wiktorowicz (2006) classified Salafis into three categories: (i) purists, who stress persuasive propagation and education; (ii) political, who emphasize the implementation

of Salafi precepts at a governmental level, and (iii) jihadis, who choose revolutionary strategies to achieve their goal. Whereas Wiktorowicz (2006) categorises Salafis mainly on the basis of strategies adopted to achieve their goals, Duderija (2007) describes Salafis on the basis of their methodology of interpretation to extract meaning from Qur'an and *hadiths* (Prophetic Traditions). According to Duderija (2011), Salafism is a modern phenomenon, but its followers employ a traditional *manhaj* (methodology) to uncover the Qur'anic and *hadiths* meaning.

In the Indonesian context, it is important to consider the factors that make Salafism appealing among Muslims. In historical terms, Indonesian Salafism proliferated a long time ago. *Persatuan Islam* (Islamic Union), *Muhammadiyah*, and the *Paderi* movement are strongly influenced by Salafism. *Dewan Dakwah Islamiyah Indonesia* (Indonesian Islamic Propagation Council) is also closely related to Saudi Arabia, with its goal being to compete with Christian missionary activity (Bruinessen 2004). All of these movements emerged in the frame of Islamic revival influenced by the interaction between tradition and modernity, international influences as well as political tensions. The Salafi movement explored in this article is not related to the general trends mentioned above, but to the new current, referred to as 'the new Salafi', which came to Indonesia in three ways: education, mainly emanating from Saudi Arabia and Yemen; human movement around in international caravans; internet and book publications (Fealy 2005).

The new Indonesian Salafis have been studied by numerous scholars. Syarifuddin (2012), a leading religious preacher in Aceh, discusses a specific topic related to the response of Salafism to the school of jurisprudence (*madzhab*). This is a significant topic due to the strong rejection by Salafism to joining any *madzhab* in Islam. According to the Salafis, joining a *madzhab* is illegal. Instead, all Muslims should refer directly to the Qur'an and *sunnah*. Syarifuddin (2012), on the basis of his debate with a group of Salafi *ustadz* (Islamic religious preachers) in a mosque in Medan, shows the weakness of the Salafi position and its preference to only follow the Salafi *ustadz*.

Studying the suicide bombings in Indonesia, particularly in Java, there were different reactions according to the Salafi subgroup: there were those who supported it, represented by the Jihadi Salafi, and those who totally rejected it, represented by the Salafi Wahhabists. The latter group argued that Islam does not allow its followers to commit suicide regardless of the reason (Rusli 2014). Another scholar, Jahroni (2013) focuses on the religious character of the relationship between Indonesia and Saudi Arabia, in which Salafism is the main element in maintaining the Saudi influence in Indonesian Muslim society. He argues that the Islamic and Arabic College of Indonesia (LIPIA) in Jakarta, which is fully funded by the Saudi government, can be regarded as the main agency to strengthen the presence of Saudi religious authority.

Hasan (2006) discussed specifically the hard-line Salafi represented by *Laskar Jihad* (the Troops of Jihad), arguing that the emergence of *Laskar Jihad* in Indonesian society, particularly in Java, is an assertion of their identity. The socio-political atmosphere in 1998, when the tension between Muslims and Christians was at its peak, provided a fertile ground for the hard-liners to make their presence felt.

Unlike Hasan, Wahid (2014) focuses his research on the role of Salafi *pesantren* (Islamic boarding schools) in Java. He sees these as playing a crucial role in training a young generation of Indonesians to be capable of teaching the Salafi *manhaj* (path or methodology). Wahid emphasises the importance of the Salafi path as it is considered the third Islamic source, after Qur'an and *hadiths*, reflecting the belief and practices exemplified by the first three generations of Muslims. It is this *manhaj* which distinguishes Salafism from other reformist movements in Indonesia, such as *Muhammadiyah* and *al Irsyad* (Wahid 2014).

Nisa (2012) discusses another aspect of Indonesian Salafism: the status of women and how they interact and practice their religion with special reference to the wearing of the *niqab* (veil). According to Nisa, wearing the *niqab* among female Salafi students is part of fulfilling Islamic obligations in order to be able to be a better believer.

#### 4. How Does a Muslim Come to Feel Deprived?

The theory of relative deprivation has been used by scholars to explain the causes and development of social movements in which an individual or a group of people feel deprived in the cultural and socio-economic spheres (Ali 2012; Morrison 1971). Agbiboa (2013), for example, claims that the emergence of Boko Haram in Nigeria is caused by socio-economic and political issues resulting from elite corruption and the prevalence of poverty among people. Similarly, Dein and Barlow (1999) argue that the underlying reason for people to join the Hare Krishna Movement in London is a feeling of existential deprivation. From this explanation, it can be assumed that an individual can feel deprived in a religious or spiritual sense, reflected in feelings of emptiness, anxiety and experiencing life with little or no meaning.

Smith and Pettigrew (2015, p. 2) define relative deprivation as "a judgment that one or one's in-group is disadvantaged compared to a relevant reference, which consequently invokes feelings of anger, resentment and entitlement". Relative deprivation itself is referred to as "a sense of deprivation" which involves comparisons with the "reference group" (Runciman 1966, pp. 10–11). The basic idea of relative deprivation is that the feeling of being deprived or dissatisfied largely depends on what someone desires to possess. This desire comes about as a result of comparing with the referent group (Morrison 1971; Runciman 1966; Webber 2007). The comparison may be carried out by an individual, which is referred to as "individual relative deprivation," or by a group, referred to as "group relative deprivation" (Runciman 1966, p. 11). Runciman further explains that the sense of deprivation may vary from one to another in terms of its magnitude, frequency or degree.

Among the Salafis, when an individual is seen to be deprived of their religious roots, his or her condition is referred to as *jahiliya* (ignorance), indicating that he or she is "unguided" and had taken the wrong path. As a corollary of this, when someone undergoes proselytization and joins the Salafis, he or she is called *muallaf* (newly converted to Islam). The use of the word *muallaf* among the Salafis implies that an individual is not a Muslim, or not a true Muslim, before his or her adaptation to Salafism. As a result of this, any knowledge of Islamic doctrines and teachings acquired before becoming a Salafi is no longer applicable, because it is impure and not in line with the understanding of *al-salaf al-salih*. If an individual experiences deprivation in the religious sense, there can be two possible responses: either returning to the religion or abandoning it. The idea behind Islamic revivalism is that Muslims return to their religion with a new and heightened commitment. In popular jargon, this is expressed as returning to Qur'an and *sunnah*.

How does a Muslim arrive at a state of feeling "deprived"? Runciman elaborates that there are three possible factors involved: education, social class and power. He elaborates that this can emerge in an individual through the process of comparison with external groups.

The basic idea discussed by Runciman is that the feeling of deprivation appears after conducting a comparison between one's lacking with another's surplus. In this comparative process, there are two aspects that should be explained, the subjects and the objects of the comparison. The latter is also called a reference group, which refers to an individual, a group or an abstract idea (Runciman 1966). According to Runciman (1966, pp. 11–15), there are three types of such groups: "comparative reference group, normative, and membership group". The first is the group an individual compares the material possession he or she has with what others possess, which can lead to a sense of lacking when found to be worse off than the other person or group. The normative group serves as the barometer of the value, and the membership group is the group to which someone feels belonging and the people with whom he or she is associated.

In the context of Islamic revivalism, the comparison conducted by some Muslims is not a linear process, but rather involves a series of thoughts and reflections of their own lives and conditions and measuring against non-Muslims in the West who appear to have better conditions than him or her. The socio, economic and political conditions of contemporary Muslim societies are in multi-dimensional crises (Dekmejian 1988). On the macro level,

Muslims are encountering crises of legitimacy and cultural, political and military conflict. These crises have driven many Muslims to seek a solution to their collective problems. Some see how their current unfortunate situation contrasts markedly with their perceived ideal of glory in the past at the time of the Prophet Muhammad, his companions and *al-salaf al-salih.* From this, they conclude that if they, as Muslims, commit to the teaching of Qur'an and *sunnah*, as their predecessors did, they will achieve glory with the reemergence of Islam as a dominant force in the world with the Qur'an and *sunnah* fully implemented in their daily lives.

Why do some Muslims seek solutions for the problems they are facing today by returning to the Qur'an and *sunnah*? The past plays a vital role in the life of Muslims who in many ways seek to find legitimacy from the past to justify what they practice today. These multiple comparative processes allow these Muslims to find the normative gap between their current behaviour and the way people behave in the past, particularly during the first three generations after the demise of the Prophet. It is this difference that triggers the feeling of being deprived; the feeling that they are no longer in line with the guidance of the Prophet.

## 5. Deprived Muslims: In the Quest for Meaning

While conducting the fieldwork, a Salafi friend in Pekanbaru invited me to see some young people to discuss a certain matter about Islam and Salafism. They were united by the same factor: searching for a solution to their existential deprivation. One of them, Ari (2015), was taking his first step towards joining the Salafi movement, and he had already attended various sermons in the Raudlatul Jannah mosque. His main reason to 'return' to Islam, he said, was that he had been married for 6 years but had no children. When he was conducting *umra* (non-obligatory pilgrimage to Makka) he prayed near the *Ka'ba* and asked God to give him children. Surprisingly, a couple of months after coming back from the *umra*, his wife fell pregnant, and finally, he had a child. This experience confirmed his decision to return to Islam, which he had neglected for a long period of time. Through Salafism, he found "true" Islam.

According to Fauzi (2015), the Salafi movement, with all its religious activities at the Raudlatul Jannah Mosque, was a place that offered support and answers to many deprived Muslims. This led Fauzi to get involved in the Salafi movement. Among the Salafi activists with whom I interacted, I have chosen three informants to represent the case studies of this study. The trio represents different social classes, professions and types of experiences of deprivation. Fauzi suffered from existential deprivation, whereas in the case of Eka, as a drug dealer, he suffered from both existential and social relative deprivation, and for Edi, both economic and existential relative deprivation. The purpose of these case studies is to show that although the informants suffered from different types of deprivation, they all turned to Islam as a solution to their problems. Indeed, as with many other 'lapsed' Muslims, Salafism was their choice among various Islamic organizations and schools of jurisprudence. The following outlines the cases of Fauzi, Eka and Edi respectively.

### *5.1. Case Study One*

Fauzi, a 40-year-old man, had been a civil servant at the Office of the Mayor in Pekanbaru since completing his undergraduate degree in Economics. As a civil servant with a position close to the governor and the mayor, he was at the centre of power. This meant he was involved in many underhand deals, with collusion and conflict among his staff making his work environment uncomfortable. His life was strongly materially oriented, and he would easily become angry if he was unable to get what he wanted. For example, he would suffer from migraines if he wanted to buy a new car but could not afford to do so and would resort to illegal means to obtain the object of his desire. For many years he chose not to spend weekends at home with his family in Pekanbaru, but rather with his boss, either in Batam or Singapore, both well-known for their glamour and luxurious lifestyle. He felt that he had achieved and experienced 'everything' in life except death.

Fauzi's first encounter with Salafism occurred when he went for *umra* to Makka in 2007, guided by *Ustadz* Armen, an influential Salafi figure in Pekanbaru. He shared a room with Armen during their stay there and Fauzi was deeply moved by the simple guidance and teaching of this preacher. After returning home, he attended Armen's sermons in various mosques, but his commitment to Salafism was short-lived and he returned to his previous lifestyle.

The turning point for Fauzi came one night in 2014 when he was at home:

> I stared at my sleeping son and suddenly began to cry because I realised I had experienced everything in life except death and I did not want my son to follow in my footsteps. The only pious friend I have is Pai. Pai told me that if I was not capable of providing a good example for my son, my son would not obey me. So, since that moment, I began to repent and learn the basics of Islamic teachings from many daurohs (religious workshops) conducted by the Salafis so that I could teach my son religious basics. I didn't want him to be a civil servant, because they are morally broken. It would be better for him to be a business person or farmer. (Fauzi 2015)

However, before joining the Salafi movement, he discussed the matter with a close friend. The friend had graduated from an Islamic boarding school of the Salafi tradition, and was able to reassure him as to its credibility. From that time on, Fauzi began attending religious sermons and informal teaching at the Raudlatul Jannah mosque, eventually finding what he was looking for. Fauzi then demonstrated his commitment to Salafism by celebrating the *aqiqa* (an Islamic ritual to welcome, protect and purify a baby) of his newborn baby at the Raudlatul Jannah Salafi mosque. The most important impact that joining Salafism has had on his life is his feeling that he is getting closer to God; he no longer feels anxious and no longer has an unhappy life. He stated that he had become a better husband and father to his children. Furthermore, his trust in God had strengthened, as had his belief that God has managed all things for him (Fauzi 2015).

Fauzi further explained what made him aware of the importance of religion as a guide in his life:

> I have had enough adventures in this mundane journey on earth. I don't need any more. All I need now is to worship God and expect a good life in the hereafter. Yes, I was disappointed with life before, because I was betrayed by others. Now, if I want something, I just ask God, not a human being. I have observed that those who worship God regularly have a pleasant life and face no hardship. (Fauzi 2015)

Continuing to compare his way of life before joining the Salafi movement and life as a Salafi activist, Fauzi (2015) recounted that:

> Before joining the Salafis, I was very engaged in earning money, whereas the people in the RJ mosque have a completely different outlook on life. They focus on giving, providing *sedekah* (optional charity). Since I joined the Salafis, I have felt calm, and no longer worry about life. This change has surprised my wife.

At the time of the interview, Fauzi had been an active Salafi in the Raudlatul Jannah Mosque for one year, having undergone a major psychological change. He narrated:

> I felt the change in my life took place during Ramadhan 2015. I never miss praying and fasting and was consistently encouraged by the ustadz to do good which made me feel good. When *Ied al Fitr* (Muslim festivity to celebrate the end of Ramadan) came, I felt a sense of calm with no desire to buy new clothes. I had also no desire to compete materially with others, something which had always been important to me in the past. (Fauzi 2015)

Despite the relatively short time that new Salafism has been in Pekanbaru compared to *Nahdlatul Ulama*, *Muhammadiyah* and some other Muslim organizations, Fauzi related to this movement for several reasons. He was deeply moved by how the Salafi preachers referred to *kitab* (religious books), using simple and understandable language:

> The sermons are heart-touching ... the way the Salafi preachers deliver them is very interesting with very simple language, suitable for the layman. If their language was too complicated, it would not be understood by people like me who went to secular schools. (Fauzi 2015)

In addition to the simplicity of the language, the politically neutral stance of Salafism also appealed to Fauzi. As he explained about one particular preacher, *Ustadz* Abu Zaid:

> *Ustadz* Abu Zaid has an Islamic boarding school. The local government wants to offer him financial aid, but he has turned it down because if he accepts it, he will be considered "part of the government", and will therefore be unable to maintain a neutral political position. Abu Zubair believes that there are many other people who will be willing to offer him aid for his *pesantren* with no strings attached. (Fauzi 2015)

When talking about why he adopted Salafism, Fauzi also mentioned that the character of the Salafi preachers was a major contributing factor in attracting people to join the movement. He told me that *ustadzs* were very sincere in their *da'wa* (Islamic preaching), and showed great wisdom in dealing with the people who consult them on religious matters.

To sum up, Fauzi joined the Salafi movement because he had become disappointed with his old life and was looking for answers to guide him away from it. Salafism appealed to him more than any other Muslim group and he was attracted to the simple language and apolitical position of the Salafi preachers.

*5.2. Case Study Two*

Eka was a single, 40-year-old man, a television journalist, and a small shop owner. His work as a journalist opened up networks to many people, including dubious characters. He frankly confessed that he had abused his profession as a journalist in order to make money. For many years, he never thought about integrity and much of the knowledge he gathered was used to blackmail people. In addition, he also took drugs, such as ecstasy, and made money by selling them. Though he came from a religious family, he never prayed, and even admitted to me that he did not know how to pray.

The *hidayah* (guidance) came to him from a Salafi and guided him to visit the mosque. Eka had conducted prayer in various mosques and had listened to many sermons but had never been satisfied. The feeling of emptiness that he felt for many years was not answered until he prayed at the Raudlatul Jannah Mosque, and listened to the sermons given there. Only then did he feel calm and that his thirst had been quenched. Since attending the mosque, he became actively involved with its many charitable activities, including serving others.

Eka was particularly drawn to the sense of social cohesion he felt among the congregation at the mosque, the respect and love shown to each other, meaning no one was left without guidance or help. For Eka, this sense of acceptance had a profound psychological effect. As a former drug dealer and drug user, Eka had spent much of his life scared of being caught, especially because a convicted drug dealer in Indonesia can face life imprisonment or capital punishment. He lived in constant fear of the police, so when he found the mosque it became a safe sanctuary for him where he no longer felt under threat.

Eka (2015) recounted the background of his life:

> I am a freelance journalist for a private television company. My life is closely related to drugs. Once I got money, I bought drugs. That's the way of my life. One of my friends was a drug dealer, and then I became involved in selling drugs too. I tried to give up in 2009 but was unsuccessful. Then, I went back to Bukittinggi, my home village (*kampung*), and tried to get away from all this. While I was there, a friend of mine in Pekanbaru who sells drugs called me and asked why I was staying so long in *Bukit*.

Eka was well aware of the wrong path he had taken and wanted to change it but did not know how. His decision to stay a while in his home village was spiritually important

and was the starting point of his return to Islam. He went to an *ustadz* asking for advice on how to reorient his life. The *ustadz* there simply told him to start praying in the mosque and disconnect himself from his 'evil' friends. When he later returned to Pekanbaru, he started praying at mosques, moving from one to another, from the *musalla* (small mosque) to a grand mosque. His first visit to a mosque did not impress him. He narrated:

> My first visit to a mosque made a bad impression on me because I didn't know the procedure. When I entered the mosque, I said *assalamualaikum* (Muslim greeting) but nobody returned my greeting. They even laughed at me. (Eka 2015)

Then, one of his friends suggested he visit a Salafi mosque. Eka recounted his first experience there and his initial involvement in Salafism:

> I visited the mosque alone. Every day, after conducting *Maghrib* prayer, I sat behind the same pillar, reading a book just to pass the time, because everybody in the mosque was reciting passages from the Qur'an. After a couple of days of doing the same thing, some members of the congregation started to pay attention to me and shook my hand. Because of their friendliness, I began to feel at home in the mosque and started learning about Islam. Three months later, I had totally given up taking drugs. I chose the Raudlatul Jannah Mosque to learn about Islam for two reasons: the members of the mosque didn't discriminate against a person like me, and they always showed respect to me even though I asked them simple or silly questions. (Eka 2015)

Eka became very active in attending the sermons and events in the Raudlatul Jannah mosque. He also prayed regularly, including *tahajjud* (optional prayer conducted between midnight and dawn), and proudly showed me the wallpaper on his mobile phone, which was a picture of the Raudlatul Jannah mosque.

*5.3. Case Study Three*

Edi was a 45-year-old man, who at the time of our meeting was selling ice cream. Riding his old motorbike, he traveled around public schools within the Pekanbaru area, where students were his main customers. He wore an Arab robe and had a long beard, which made him easily identifiable as a Salafi. His first acquaintance with Salafism started when his father enrolled his younger sister in Pesantren al Furqan, the oldest Salafi *pesantren* in Pekanbaru. After that time, he was regularly asked by his father to deliver a package of food to his sister. The regular visits to his sister's *pesantren* gave him an initial experience with Salafism but this ended when he got a job as a fish supplier for a petroleum company in Rumbai, a suburb of Pekanbaru. He was kept very busy and was well paid, but he told me that for him, the money was not a *berkah* (blessing). His father was not happy about him mixing with what he regarded as 'vulgar' people in the fish market, and they grew apart. Edi felt that his life was meaningless, and he was aware that he had adopted bad habits, such as drinking alcohol. After 13 years, he resigned from the company and started selling ice cream.

> I had sufficient money, but I spent it on useless things. I then decided to give up my job and search for the real meaning of life. I started going to places where I could join informal religious teaching sessions, including Tabligh Jamaa on Sumatra Street. I just felt that my life was meaningless if I didn't learn about Islam. (Edi 2015)

However, Edi's involvement with Tabligh Jamaa, a Sunni group in Sumatra Street Pekanbaru, only lasted a few months. He recounted:

> The preaching began after *Maghrib* (dusk) (6 pm). I met some of my friends there. I am originally from Pekanbaru. After listening to several preachers, I felt that it was only indoctrination; there was no discussion or dialogue. (Edi 2015)

The turning point in his life occurred when his sister gave him some Salafi clothes: above-the-ankle trousers, a white and grey ankle-length Arab dress, and a white Islamic

headpiece. He felt comfortable wearing them and grew a beard. Psychologically, he felt calm, as if he had found something for which he had been searching for a couple of years. He remembered his past experience with Salafism and decided to become further involved in their activities in order to quench his thirst for inner peace. This led him to visit the Raudlatul Jannah Mosque every day after work to spend a couple of hours listening to religious sermons.

During his involvement with the Salafis, he realized that many of the rituals he had performed previously when praying were incorrect and the Salafis guided him onto the right path. In addition, he found that the sense of solidarity was very strong among the Salafi members. As a person of modest means, he received many forms of material and emotional support from his Salafi peers. He also felt that the friendship in the Salafi group was based on loving God. Finally, he had found his 'real home' where he could quench his spiritual thirst.

Edi's commitment to Salafism was firm, to the extent that after traveling from school to school on his motorcycle from 6 a.m. until 4 p.m., he would attend the mosque until 9 pm and then he would travel nine km back home. He continued getting up at 4 a.m. and only going to bed after 11 p.m. for several years, but still felt energized on an average of 4 h of sleep a night. He also mentioned that he never visited his local mosque ever again, once he had found Salafism.

## 6. Deprived Muslims: A Reversion

The above three case studies all have a common thread: an initial lack of spiritual direction in life with an accompanying sense of meaninglessness and dissatisfaction with other Islamic groups, concluding with finding 'the truth' of Salafism. Following on from the personal accounts given by participants, this section looks at different types of deprivation.

Glock and Stark (1965) listed five types of deprivation: economic, social, organismic, ethical and psychological, and Dein and Barlow (1999) adding a sixth: existential deprivation. Economic and social deprivation are among the most common forms of deprivation faced by those who suffer from relative deprivation (Ali 2012), and therefore, these two types are explored in this study; while the three others (organismic, ethical and psychological) are not relevant to the phenomenon of Salafi development in Pekanbaru. Organismic deprivation refers to deficiencies in either physical or mental health; ethical deprivation refers to the sense that society does not offer appreciation for an individual life; and psychological deprivation refers to the lack of psychic rewards felt by an individual, such as love and affection (Stewart 2010). The emphasis on economic, social and existential deprivation will be elaborated further on the basis of perspectives and life stories of participants obtained during the fieldwork.

In practical terms, an individual can experience one or more types of deprivation in their life. For instance, being a drug dealer not only entails illegal activities which can cause estrangement with one's family or friends but also comes with inherent risks involving other suppliers and customers. This can lead to a sense of social alienation or social relative deprivation. In addition to this, in relation to religious doctrine, the money earned from selling drugs is considered illegal in Islam, and therefore, spending the money earned from this activity is also illegal and utterly condemned. Illegal money is not blessed by God and Muslims believe that anyone who uses that money will penalized by God in this world or the next or both. Eka was aware of this but chose to ignore it until the end of 2014 when a sense of alienation and feeling of emptiness he felt led him to seek a new direction in life.

The experiences of existential or spiritual deprivation felt by the informants fell into two categories: a sense of a meaningless life and dissatisfaction with the existing Islamic groups. Both of these forms led them to find the 'truth' of Salafism, and for some informants, a combination of the two categories above was experienced simultaneously. In the first category, the feeling of emptiness initially arose when one begins to question what one is doing in life, and begins to make changes. This process can be called the self-identification act. In the process of change, self-identification is an act to assess one's situation; it is a

precursor that is followed by a reaction to get out of the situation. Age is one of the factors that trigger one's awareness. Isrul (2015), for example, started to question the meaning of life when approaching 40 years of age. For him, turning 40 was a very significant point in his life. He felt that if he was unable to find the answers for which he had searching for years, he would no longer be able to improve his life to have a better spiritual condition. As a graduate of a secular university, his limited religious background hindered his ability to interpret his experience. He was well-established financially but was poor spiritually. For him, adapting Salafism totally changed his view about life: Allah became central to his identity, represented by his willingness to spend his life in the mosque, instead of in the *kampung* (native village). In a similar sense, Anwar (2015) also expressed his anxiety about life when turning forty. Although financially secure, he did not have a remedy for his emptiness until he listened to a sermon delivered by *Ustadz* Abu Zaid, a Salafi preacher, entitled: *Aku Datang Wahai Kekasih* (I am coming, oh my Love). Anwar confessed that it was this sermon that totally changed his life. Then, along with his Salafi friends, he donated a great amount of his wealth to establish *al Bayyinah* School, a leading Salafi school in Pekanbaru. He also decided to resign from his established position in the Ministry of Forestry, saying that he no longer needed a salary from the government.

In line with Anwar's case, Joko (2015) wondered how he could 'invest' his wealth for the hereafter life. Being economically secure, Joko was aware that human life in this world is very short and subsequently the opportunities to enjoy wealth are also limited. Islam offers an explanation for people about the meaning of wealth, allowing them to enjoy that wealth in this world as well as in the hereafter by using it for *da'wa* (preaching). The expected rewards of the invested *da'wa* funds have an important meaning bestowed by Islam, allowing the owners to enjoy a 'new opportunity' to get involved in *da'wa* by donating their wealth instead of giving sermons.

Unlike the informant above, Ari (2015) returned to Islam after his prayer to have children was granted by Allah. The sentiment that God had been 'present' in his life was strongly felt after doing *umra*, and this led him to get involved more deeply with religious activity, particularly in the Raudlatul Jannah mosque where regular teachings were available during the weekdays. His closeness to Allah helped him understand his destiny; he believed that it is Allah the Almighty who manages everything, and therefore, the return to Islam is a reasonable choice for those who seek a meaningful life.

The second category, dissatisfaction with the existing Islamic groups, is also part of existential deprivation. Hasan's (2016) experience is worth narrating here. He was recruited as a member of *Negara Islam Indonesia* (Islamic State of Indonesia), a clandestine movement which aims to establish an Islamic state in Indonesia, when he had been an undergraduate student. His last position in *Negara Islam Indonesia* was the Principal of the Recruitment Department for the northern part of Sumatra. His main task was to recruit people to become members of Negara Islam Indonesia, and in order to do so, a deep knowledge of Islam was required. He was thirsty for religious knowledge but *Negara Islam Indonesia*, as an organization, did not provide it. This had made him anxious. As a *Negara Islam Indonesia* member, he was indoctrinated that all Muslims are disbelievers unless they join the *Negara Islam Indonesia*. However, when he was pursuing a master's degree in the *hadith* sciences, he started to question *Negara Islam Indonesia*'s stance. He then visited Tarmizi, a Salafi preacher, to discuss various religious matters that had made him anxious. After a series of discussions with Tarmizi, he found what he needed to finally leave the *Negara Islam Indonesia*. Salafism provided him with arguments which were, according to him, 'very logical, and in line with his thoughts'. In the same way, Bilal (2015) returned to the 'right' path as a result of his dissatisfaction with liberal Islam, which could not satisfy his religious thirst. According to him, the liberal Islamic approach places an over-emphasis on reasoning instead of Qur'an and *hadiths*. In other words, this approach did not deeply touch his heart. In addition, the proponents of liberal Islam are highly influenced by Western thought, which in many cases, affects their interpretation of the religious texts. This can be seen

from the main issues upheld by this group, such as supporting the women's emancipatory movement, freedom of religion, secularism and allowing interfaith marriage.

In a similar vein, Doni (2015) and Said (2015) expressed their disagreement with the existing Islamic organizations, such as *Nahdlatul Ulama* and *Muhammadiyah*, emphasising that what Muslims need is only to imitate the *al-salaf al-salih*, the earliest and purest Muslim generations. It is clear that being a member of a certain Islamic group does not ensure that an individual will obtain satisfactory explanations about Islam, as seen in the case of Hasan and Bilal, which therefore led them to search for more trusted answers from other Islamic groups. The strength of Salafism, in this case, is its capacity to provide reliable and confidential answers, based on the Qur'an and *hadiths*, which can meet the needs of some Muslims.

In addition to dissatisfaction with the existing Islamic organizations, the perceived truth of Salafism is also an important factor that led many people to join the Salafi movement. Syukur (2015), Jafar (2015) and Habib (2015), for example, stopped their studies at the secular university due to their perception that the secular sciences they had been learning were not 'recommended' religiously, and therefore would not be useful to them in the hereafter. This awareness arose after attending the sermon in which Buya Jufri explained the superior position of Arabic and religious knowledge compared to the secular one. Following this, they decided to learn Arabic and Islamic studies at the *pesantren* instead of secular sciences at the university. They then became teachers of religious subjects at the Salafi schools. Salafism, in this case, re-oriented their life and led them to prioritize the Salafi teachings over other matters, including secular sciences taught at a university.

In a slightly different narrative, Nimra (2015) and Musa (2015) found the 'truth' of Salafism and joined it without dropping out of the university in which they studied mathematics and chemistry. In order to be able to take 'double' courses at the university and *pesantren*, they decided to live in a *pesantren* dormitory, allowing them to learn and interact with Buya Jufri intensively. By doing this, they were able to complete both their academic degree at the university and religious training in *pesantren.* They then became lecturers at the university. However, the experience of living and learning in the Salafi *pesantren* kept them active in the Salafi *da'wa.* Nimra and Musa are presently Head of the Department of Education and Chairman of *Ubudiyya* Salafi Foundation, respectively.

The interviews above show that the participants have adopted Salafism after being deprived for a certain period of time. Salafism, in this case, is a referent group for them which is regarded as the ideal standard of Islam. Runciman (1966) explains that there are three types of referent groups: group, individual or an abstract idea. In the process of adaptation to Salafism as shown in the case studies and interviews above, these three types of referent groups exist together, represented, respectively, by the Salafi preachers, the Salafi doctrine and the Salafi movement. These referent groups are interlinked, with the Salafi preachers playing the most important part. The following Table 1 shows the perception of the informants (represented by the three case studies above) on the three types of the referent groups explained above.

**Table 1.** Perception of the informants on the referent groups.

| Salafi Preachers as an Individual Referent | Salafism as an Abstract/Idea Referent | Salafi Movement as a Group Referent |
| --- | --- | --- |
| Sincere | Simple | Strong solidarity |
| Wise | Pure | Gives culture |
| Non-political | Based on Qur'an and *hadiths*, therefore authoritative | Sincerity of friendship due to love of Allah |
| Respect the layman | | |
| Expert in Islamic sciences | | |

Adaptation is a slow and long process and for some individuals, matters related to their relative deprivation have to first be resolved. Data generated from the informants identified that choosing Salafism as a resolution for their crises was mainly based on two reasons: searching for a new sense of self or having had previous experience with Salafism. Eka and Fauzi represent the former, where Eka had first listened to sermons in other mosques but found they did not address his needs, whereas the strong sense of community in the Raudlatul Jannah mosque appealed to him. In addition, the large number of people attending prayers impressed him, particularly at the *Fajr* prayer. It should be noted that in Indonesia, most Muslims are still in bed at the time of the *Fajr* prayer, whereas thousands congregate in the Raudlatul Jannah mosque regularly for the event.

In spite of his sinful past, Eka felt accepted at the Raudlatul Jannah Mosque and proudly told me that a couple of months after regularly attending the mosque, he was asked by the Salafi Foundation to join the *Qurban* (animals slaughtered during *Eid el Adha* festival) committee, and also the Salafi team to distribute aid to needy people in the countryside. For him, his previous alienation was replaced by a strong sense of community and religious experience. The Salafi preachers and the Salafi community were the two crucial referent groups that satisfied his needs. Eka (2015) sums up his perceptions of the Salafi movement. "I came in and out of numerous mosques in Pekanbaru, but none of them had strong solidarity as I had found in RJ mosque. As a layman, my questions about Islam were always answered politely."

The first informant, Fauzi, also found that his existential deprivation was resolved by Salafism, which he found was a simple and straightforward way to understand Islam on the basis of Qur'an and *sunnah*. He attributed this to the sincerity of the preachers, and the simplicity of the teachings which he felt was spoken from the heart and touched him deeply. He recounted: "I started to select the preachers; I don't want to listen to sermons other than those conveyed by the Salafi preachers." He explained further: "I found peace and sincerity in the Salafi group. They develop the culture of giving rather than receiving" (Fauzi 2015).

In a similar vein to Eka and Fauzi, Edi had felt a lack of meaning in his life, compounded by financial difficulties. In his case, he adopted Salafism as a result of his sister giving him Salafi clothes to wear, which triggered a recall process of his previous interactions with the movement. Edi confessed that his financial worries were largely resolved by joining the Salafis since they gave him regular payment for his children's education in Salafi schools and two lump sums a year towards other expenses. In addition, when he followed the Salafi commitment of fasting two days a week, on Mondays and Thursdays, the Salafis provided the food when he broke his fast. He narrated:

> I have been active in the RJ Mosque for many years. In spite of my economic situation, which is economically unlucky, I have never been insulted or discriminated against by others. That is why I am so comfortable within the Salafi group. Though I sell ice cream all day, I never feel tired when I arrive at the Raudlatul Jannah Mosque. The feeling of tiredness usually comes when I arrive home. (Edi 2015)

In addition to his financial needs being met, from a religious perspective, Edi felt that he had become aware of the false practice of prayer he had conducted for many years, which the Salafis claimed was not in line with the teachings of the Prophet. He stated that Salafism had taught him the correct way to worship God.

Table 2 outlines the mental state of the informants prior to embracing Salafism, comparing the three referent groups.

Table 2 above shows that the informants experienced existential and social relative deprivation before they felt their needs were met when they adopted Salafism. The comparison they made with the referent groups, the individual Salafi, the doctrine of Salafism and the Salafi community can be viewed as the initial identification they experienced prior to seeing their problem resolved.

**Table 2.** Mental state of the informants and comparison with referent groups.

| Informant | Informant's Mental State | State of the Referent Group within the Salafi Group | | |
| --- | --- | --- | --- | --- |
| | | **Individual** | **Idea** | **Group** |
| Fauzi | • Consumed material pleasure<br>• Feelings of anxiety<br>• Being a bad Muslim | • Sincerity of Salafi preacher | • Purity of Salafism and authority of its teaching | • Salafi culture rising above material culture |
| Eka | • Never prayed<br>• Involved in criminal activities, including consuming and selling drugs<br>• Felt anxious and alienated | • The sense of calm and sincerity of Salafi followers | • The simplicity of Salafi teaching | • The high attendance at Salafi mosques<br>• Sincerity of friendship among the Salafis |
| Edi | • Relationship with his father not warm<br>• Felt not blessed by God<br>• Lack of meaning in life | • The Salafis only consume *halal* food<br>• The wisdom and sincerity of the Salafi preacher | • The purity and authority of Salafi teaching | • Strong sense of solidarity (*ukhuwwah*)<br>• The Salafis helping each other in God's name |

Among the three referent groups, the Salafi preachers, as the individual referents play an important role, because they function as a Salafi window through which the newcomer can see 'inside' Salafism. Fauzi was strongly influenced by the humble life of *Ustadz* Armen, the Salafi preacher, who taught him that Islam is quite simple. Similarly, Edi and Eka were impressed both by the extensive knowledge and humble life of *ustadz* Bukhari. Salafism, as taught by these preachers is considered simple, pure and therefore authoritative. The Salafi preacher's interpretation of Islam is then implemented by Salafi laymen. The only reference to the Salafi preachers then creates homogeneity within the group. It is clear that the Salafi preachers have played a significant role in attracting people to join the Salafi movement.

From the above accounts given by informants concerning their transformation, Salafism appears to offer some positive advantages, such as relief from anxiety, creating social solidarity, and satisfying their cognitive needs through the explanations given by the Salafi preachers. When describing the process of reversion, all informants use the word *hijra* (migration). The underlying reason which motivates them is that they feel they are doing *hijra* (spiritual migration) under God's *hidaya* (guidance). Proponents of Salafism describe the condition before transformation, as being one of *jahiliyya*. *Jahiliyya* is a situation where an individual is deprived of religious experience. The process of moving from being unguided to guided is called *hijra*. The new Salafi recruit is called a *muallaf*. Table 3 below explains the relationship of these four sequences, *jahiliyya, hidaya, hijra, muallaf*.

**Table 3.** Sequences of adaptation to Salafism and its equivalent condition according to relative deprivation theory.

| *Jahiliyya* | *Hidaya* | *Hijra* | *Muallaf* |
| --- | --- | --- | --- |
| Existential deprivation:<br>• Non-practicing Muslim<br>• Practicing but still doing illegal rituals<br>• Practicing rituals but does not understand the *dalil* (argument) | Assessment/act of identification then finding a solution:<br>• Being conscious of one's Muslim identity<br>• Being aware of positive advantages of Islam | Turning point:<br>• From *bid'a* (innovation) to *sunnah*<br>• Moving from an unconscious state of one's Muslim identity to a fully conscious one | New-born Muslim:<br>• Know the *dalil*<br>• Imitate the Prophet in all aspects of life |

Before discussing the Salafis and the factors that draw people to choose it, it is worth looking at the factors that make Islam attractive to those suffering from a sense of deprivation.

## 7. Relative Deprivation and the Return to Religion

It is commonly believed among those who support the socio-economic deprivation theory that people's reversion, or return to religion, is as a result of the feeling of material deprivation (Wimberley 1984; Durkheim 1951; Weber 1946). Their hypothesis is that people are subjected to one or more types of deprivation, such as socio-economic, and will strive to find a solution to their dissatisfied social and material conditions (Glock 1964; Runciman 1966; Wimberley 1984). Stark and Smith (2010), for example, cited some scholarly observations on the underlying reasons that drive people to join Pentecostal Protestantism in Latin America: poverty, illiteracy and health problems. Some other scholars, such as Berger (1969), Davidson (1977) and Stark and Bainbridge (1980) also considered that the socio-economic deprivation proposition led those who are economically unlucky to religiosity. The term religiosity here refers to a complex combination of cognitive and behavioural tendencies, such as a belief in God, rituals and different states of consciousness.

On the basis of the deprivation theory, religion is perceived as being capable of alleviating people's suffering by providing explanations that 'refresh' their understanding of life and its meaning. In other words, it can be said that religion serves as a compensator (Stark and Bainbridge 1980). The capacity of religion to provide profound meaning to life can be used by materially deprived people as a justifiable excuse to solve their problems. However, Stark and Smith (2010) explained that the proselytization of underprivileged people to Pentecostal religious groups in Latin America were not caused by material deprivation, but rather influenced by social ties and spiritual satisfactions. The absence of the rich people in those groups is only because they do not appeal to the rich. Spiritual satisfaction should be highlighted here as the most important goal pursued by people in religions, therefore, deprivation theory should not be confined to material aspects only, but extended to include religious or spiritual deprivation (Stark and Smith 2010).

The gist of spiritual or religious deprivation theory is that people will look to supernatural solutions to overcome their dissatisfied existential and moral conditions. This hypothesis is relevant to explain the underlying reason for the process of transformation in Pekanbaru Salafism, in which most of its members are financially secure but religiously unhappy. Instead of pursuing material rewards from their transformation, they seek spiritual fulfilment provided by Salafism. They reach that fulfilment through learning religious teachings and maximising the use of their wealth for religious purposes, such as helping the needy and constructing religious buildings.

In general, there are various reasons why people tend to return to religion at a time of deprivation. Geertz (1973), for example, stated that religion is a universal element of human cultures due to its ability to satisfy human spiritual needs; it is also considered as the most powerful force in human society that influences relationships with each other (Mc Guire 2002). The return to religion, in this case, is a reasonable act due to its potential to solve the problem of deprivation.

As a universal element, religion has been defined in many ways, the most popular of which is through substantive and functional approaches (Mc Guire 2002). The former defines religion on the basis of its essence: what qualifies something as religion or not a religion, while the latter focus on its social function (Mc Guire 2002). The two approaches above seem to distinguish between the essence of religion and its function. Practically, religion cannot be separated in a clear-cut sense into these two categories because it could socially function due to its essence which provides an explanation of the mystery of life and offers a profound meaning of life for its followers. Both approaches are accordingly intertwined and therefore, the role of Salafism, for instance, can be understood in terms of its capability to provide the certainty needed by the anxious and curious members; and this certainty originates from its deep explanatory essence about all aspects of life.

The way religion offers meaning about life is by providing explanations of situations, experience and events (Mc Guire 2002). For example, in Islam, different states of human life, with some people poor and others rich, are given a meaning when it is interpreted as God's will. The Salafis believe that Allah has already managed everything in the world, and therefore, complaining about poverty is understood as complaining about Allah's will, which is not allowed in Islam. Total surrender to Allah's will is regarded as the highest level of sincerity that a Muslim can achieve. Adopting such an interpretation can prevent the 'unlucky' from being resentful towards the rich on one hand, and on the other hand, ensure the wealthy that all their property belongs to God and should be used in legitimate ways. Linking explanations of different fates to God's will offers a clearer perspective that makes people calm psychologically and helps give their life meaning.

The role of religion in providing meaning for people is significant as the meaning itself is bestowed, not intrinsically found in it (Berger 1969). As a result, it is commonly found that similar phenomena are understood in different ways depending on the strength of one's attachment to religion. For example, a tragedy faced by a deprived Muslim could be understood as a burden in their life, while a committed Muslim may have a different outlook by perceiving it as God's will to elevate him or her to a higher degree. The teaching that Allah will not let a Muslim upgrade his religiosity without giving him or her comprehensive and continuous trial in his life is clearly mentioned in the Qur'an. In this sense, a total submission to God's will has led people to understand life challenges in a positive way: to improve their religious understanding and practice, which eventually can affect their socio-economic conditions.

## 8. Conclusions

This article contends that those who return to Islam have suffered from a sense of relative deprivation, particularly existential deprivation. Being relatively deprived leads those non-religious Muslims to seek the truth which, in the cases presented above, is influenced significantly by the Salafi community, preachers and doctrines. In light of the Islamic revivalism phenomenon, the feeling of relative deprivation is presumed to be closely related to the negative impacts of modern life. In the case of Pekanbaru, this can be observed in individuals who report a lack of meaning and being disoriented in life, or being involved in religiously prohibited deeds. Moreover, dissatisfaction with the existing Islamic groups due to their perception that these kinds of Islamic practices are not pure and are less authoritative led some of the informants to leave and join Salafism, which is perceived as being more authoritative and genuine.

**Funding:** This research was funded by Australia Award Scholarship, grant number ST000K7C9.

**Institutional Review Board Statement:** The study was conducted in accordance with the Declaration of Helsinki, and approved by the Ethics Committee of Western Sydney University (protocol code H 11195, approved on 16 July 2015).

**Informed Consent Statement:** Informed consent was obtained from all subjects involved in the study.

**Data Availability Statement:** Not applicable.

**Conflicts of Interest:** The author declares no conflict of interest.

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
