# Peer review of "Deprived Muslims and Salafism: An Ethnographic Study of the Salafi Movement in Pekanbaru, Indonesia†"

_religions, doi:10.3390/rel13100911_

Round 1

Reviewer 1 Report

While the paper has managed to present the process of reversion to Salafism in part of the three deprived Muslims who have returned to Islam as a solution to their sense of deprivation, the objective of the research is still not clear. Likewise, the paper has not been able to show its significance and importance. The research should not be assertive of the role of Salafism only by looking at three case studies. 

The conclusion also is very weak and too short. It needs elaboration, concrete substance and even analysis and recommendations to end the paper. 

All these need to be addressed before the paper can be published.

Author Response

Response to Reviewer 1 comments

Point 1: While the paper has managed to present the process of reversion to Salafism in part of the three deprived Muslims who have returned to Islam as a solution to their sense of deprivation, the objective of the research is still not clear.

Response 1: The objective of this study is to explore the role of Salafism in solving the problem of existential deprivation among non-religious Malays which finally lead them to convert to Salafism and join the Salafi community of Pekanbaru. Exploration of these aims will reveal the role Salafism plays in providing the solution for deprived non-religious Malays, experiencing some negative impacts of modern life and deviated Islamic beliefs and practices.

Point 2: . Likewise, the paper has not been able to show its significance and importance

Response 2: The main aim of this article is, therefore, to explore the phenomenon of Islamic revivalism represented by the Salafi Movement in Pekanbaru, Sumatra in the term of how it finds its role within the community by solving relative deprivation suffered by some non-religious Muslims. Study on the Sumatran Salafism is required as most recent studies on Indonesian Salafism have focused on Java, with scant reference to other regions of Indonesia.

Point 3: The conclusion also is very weak and too short. It needs elaboration, concrete substance and even analysis and recommendations to end the paper.

Response 3: This article contends that those who return to Islam have suffered from a sense of relative deprivation, particularly existential deprivation. Being relatively deprived leads those non-religious Muslims to seek the truth which, in the cases presented above, is influenced significantly by the Salafi community, preachers and doctrines. In the light of the Islamic revivalism phenomenon, the feeling of relative deprivation is presumed to be closely related to negative impacts of modern life. In the case of Pekanbaru, this can be seen in feelings of lacking meaning and being disoriented in life, or being involved in religiously prohibited deeds. Moreover, dissatisfaction with the existing Islamic groups due to their perception that these kinds of Islam are not pure and are less authoritative led some of the informants to leave and join Salafism, which is perceived as being more authoritative and genuine. The case studies presented show that the return of those non-religious Muslims to Islam suggests that they obtain positive advantages for their spiritual and cognitive needs provided by Islam, i.e. Salafism. Finally, it can be said that Salafism, as the representation of the Islamic revivalism phenomenon in Pekanbaru, plays an important role in providing a solution for the middle aged Muslims who experience existential relative deprivation. Their return to Islam suggests that they obtain positive advantages for their spiritual and cognitive needs provided by Islam, i.e. Salafism.

Reviewer 2 Report

  1. Reversion to salafism is unclear. Salafism is an extreme form of Sunni Islam. How is adhering to it reversion? Hijra means in the salafi context the interruption of the non-Islamic way of life. The article should reconsider salafism as radical change (and not reversion) within one’s religious and social lives.
  2. The same can be said about the notion of deprivation. It is a judgment value and not an observable reality. The use of reversion and deprivation undermine the scholarly quality of this article. Economic and social deprivation can be studied sociologically, but spiritual deprivation is a psychological topic, and the author is not a psychologist. Perhaps this is the reason the method is flawed.
  3. Once the cases are explored it appears the interviewed embrace salafism because of various reasons, and not only because of spiritual deprivation.
  4. Salafism in the West is irrelevant to salafism in Indonesia. The author should delete the elements about global salafism, Bosnia and Sweden, and expand rather on the history and presence of salafism in Indonesia, using sources in Indonesian language. This section is poorly written and needs to be thoroughly developed.
  5. The conclusion is too short and weak.
  6. The article needs to be submitted to a native speaker of English. It cannot be published as such.
  7. The author should reconsider deprivation as a framework.

Author Response

Response to Reviewer 2 comments

Point 1: Reversion to salafism is unclear. Salafism is an extreme form of Sunni Islam. How is adhering to it reversion? Hijra means in the salafi context the interruption of the non-Islamic way of life. The article should reconsider salafism as radical change (and not reversion) within one’s religious and social lives. Response 1: I have decided to use conversion instead of reversion. This is due to their strong rejection to the period before becoming a Salafi. New method paragraph has been added to support this (in red)

Point 2: Once the cases are explored it appears the interviewed embrace salafism because of various reasons, and not only because of spiritual deprivation.

Response 2: in red

Point 3: Salafism in the West is irrelevant to salafism in Indonesia. The author should delete the elements about global salafism, Bosnia and Sweden, and expand rather on the history and presence of salafism in Indonesia, using sources in Indonesian language. This section is poorly written and needs to be thoroughly developed.

Response 3: the element was deleted

Point 3: The conclusion is too short and weak.

Response : This article contends that those who return to Islam have suffered from a sense of relative deprivation, particularly existential deprivation. Being relatively deprived leads those non-religious Muslims to seek the truth which, in the cases presented above, is influenced significantly by the Salafi community, preachers and doctrines. In the light of the Islamic revivalism phenomenon, the feeling of relative deprivation is presumed to be closely related to negative impacts of modern life. In the case of Pekanbaru, this can be seen in feelings of lacking meaning and being disoriented in life, or being involved in religiously prohibited deeds. Moreover, dissatisfaction with the existing Islamic groups due to their perception that these kinds of Islam are not pure and are less authoritative led some of the informants to leave and join Salafism, which is perceived as being more authoritative and genuine. The case studies presented show that the return of those non-religious Muslims to Islam suggests that they obtain positive advantages for their spiritual and cognitive needs provided by Islam, i.e. Salafism. Finally, it can be said that Salafism, as the representation of the Islamic revivalism phenomenon in Pekanbaru, plays an important role in providing a solution for the middle aged Muslims who experience existential relative deprivation. Their return to Islam suggests that they obtain positive advantages for their spiritual and cognitive needs provided by Islam, i.e. Salafism.

Point 4: The article needs to be submitted to a native speaker of English. It cannot be published as such.

Response 4: The article was edited by a native speaker of English.

Point 5: The author should reconsider deprivation as a framework.

Response 5: New section on the deprivation framework has been added (in red) discussing it in more detail.

Round 2

Reviewer 2 Report

  1. The author uses revert seven times in his second version of the text. This should be replaced by convert. This should be corrected.
  2. Deprivation: the additions of the author about the deprivation hypothesis leave me with a mixed opinion. The basic premise that Muslims in Indonesia who “convert to salafism” were deprived (that is lacked basic and cognitive spiritual needs) is a normative statement that cannot be measured. People obtain their spiritual and cognitive needs from various sources, and not necessarily from religion, especially in the modern times. The study would be still publishable without the deprivation framework. I leave the decision to publish or not to the editorial board.
  3. If the author decides to maintain the deprivation framework, then he has to delete the normative tone in the text which unwillingly presents the author as supporting salafism as a solution to “deprivation”. Here is an example:
    1. The case studies presented show that the return of those non-religious Muslims to Islam suggests that they obtain positive advantages for their spiritual and cognitive needs provided by Islam, i.e. Salafism. Finally, it can be said that Salafism, as the representation of the Islamic revivalism phenomenon in Pekanbaru, plays an important role in providing a solution for the middle aged Muslims who experience existential relative deprivation. Their return to Islam suggests that  they obtain positive advantages for their spiritual and cognitive needs provided by Islam, i.e. Salafism.

  1. There are still mistakes in English. Here are three examples:

    1.  On page 1: Sumatra in the term of how should be „in terms of”
    2. On page 12: to interpret his experience of his should to interpret his experience.
    3. On page 12: had searching for for years should be had searching for years.

Author Response

  1. The author uses revert seven times in his second version of the text. This should be replaced by convert. This should be corrected.

     Response: The study of this Salafi Group is part of the effort to understand the religious change in Pekanbaru. The change occurs at two levels: firstly, within a given religious tradition (reversion), and secondly between religions, (conversion). The former is manifested in the change of religious affiliation, preferences and participation, while the latter may involve converting from one religion to another, for example Christianity to Islam or vice versa. Since the context of this study is the Muslim society of Pekanbaru, as a result of the development of Salafism, my focus is on reversion. Moreover, the use of the concept of conversion to refer to religious change within one religion is uncommon in scholarly literature. Reversion is therefore more relevant to be used.

2.   Deprivation: the additions of the author about the deprivation hypothesis leave me with a mixed opinion. The basic premise that Muslims in Indonesia who “convert to salafism” were deprived (that is lacked basic and cognitive spiritual needs) is a normative statement that cannot be measured. People obtain their spiritual and cognitive needs from various sources, and not necessarily from religion, especially in the modern times. The study would be still publishable without the deprivation framework. I leave the decision to publish or not to the editorial board.

3. If the author decides to maintain the deprivation framework, then he has to delete the normative tone in the text which unwillingly presents the author as supporting salafism as a solution to “deprivation”. Here is an example:

    1. The case studies presented show that the return of those non-religious Muslims to Islam suggests that they obtain positive advantages for their spiritual and cognitive needs provided by Islam, i.e. Salafism. Finally, it can be said that Salafism, as the representation of the Islamic revivalism phenomenon in Pekanbaru, plays an important role in providing a solution for the middle aged Muslims who experience existential relative deprivation. Their return to Islam suggests that  they obtain positive advantages for their spiritual and cognitive needs provided by Islam, i.e. Salafism.

                 Response: deleted

4. There are still mistakes in English. Here are three examples:  

    1.  On page 1: Sumatra in the term of how should be „in terms of”
    2. On page 12: to interpret his experience of his should to interpret his experience.
    3. On page 12: had searching for for years should be had searching for years.

Response: done